# Investigation of Glyphosate Removal from Aqueous Solutions Using Fenton-like System Based on Calcium Peroxide

**Fan Li** [1], **Thomas Shean Yaw Choong** [1,2,*], **Soroush Soltani** [1], **Luqman Chuah Abdullah** [1,2], **Siti Nurul Ain Md. Jamil** [3,4] and **Nurul Nazihah Amerhaider Nuar** [3]

1   Department of Chemical and Environmental Engineering, Faculty of Engineering, Universiti Putra Malaysia, Serdang 43400, Selangor, Malaysia
2   Institute of Tropical Forestry and Forest Product (INTROP), Universiti Putra Malaysia, Serdang 43400, Selangor, Malaysia
3   Department of Chemistry, Faculty of Science, Universiti Putra Malaysia, Serdang 43400, Selangor, Malaysia
4   Centre of Foundation Studies for Agricultural Science, Universiti Putra Malaysia, Serdang 43400, Selangor, Malaysia
*   Correspondence: csthomas@upm.edu.my

**Abstract:** Glyphosate [N-(phosphonomethyl)-glycine], an organophosphate broad-spectrum herbicide, is a serious environmental contaminant that poses a significant threat to humans. It can be detected as a contaminant in water; thus, effective procedures for its removal are urgently required. The present study investigated the performance of glyphosate removal from aqueous solutions using a Fenton-like system. Calcium peroxide ($CaO_2$) was used as a source of hydroxyl free radicals with $Fe^{2+}$ as a catalyst to trigger the Fenton reaction. Fourier transform infrared spectroscopy (FTIR), x-ray diffraction (XRD), and potassium permanganate titration were carried out for characterization of calcium peroxide. The effect of operating parameters such as pH, the molar ratio of $Ca^{2+}:Fe^{2+}$, the initial dosage of calcium peroxide, and the initial glyphosate concentration on the removal efficiency was studied, respectively. The maximum total phosphorus (TP) and chemical oxygen demand (COD) removal were 94.50% and 68.60%, respectively, within 120 min under optimal conditions (pH = 3.0, initial glyphosate concentration = 50 ppm, initial $CaO_2$ dosage = 0.5 g, molar ratio of $Ca^{2+}:Fe^{2+}$ = 6, room temperature). Experimental data were analyzed using zero-order, first-order, second-order, and Behnajady, Modirshahla, and Ghanbary (BMG) kinetics models. The degradation kinetics of glyphosate could be mostly fitted with the BMG kinetics model. This study demonstrated that calcium peroxide could be considered an effective oxidant for glyphosate wastewater treatment.

**Keywords:** Fenton-like system; calcium peroxide; herbicide; kinetic study

## 1. Introduction

Water contamination, originating from industrial or agricultural activities or domestic wastewater, is a major problem with environmental, health, social, and economic consequences. Therefore, it is important to emphasize the negative impact of agricultural wastewater generated by the widespread use of chemical fertilizers, pesticides, and herbicides. [1]. Owing to its high concentration, high salinity, high total phosphorus content, and high COD value, agricultural wastewater must be effectively treated before being discharged into the environment [2]. Glyphosate (N- (phosphonomethyl) glycine, $C_3H_8NO_5P$, PMG; chemical structure shown in Figure 1) is the active ingredient of Roundup, marketed as a non-selective, post-emergence, broad-spectrum organophosphate herbicide which is used globally in agriculture, forestry, and cities [3]. With the development of weed resistance, glyphosate consumption increased from 16 million kg in 1994 to 79 million kg in 2014 [4]. Considering the harm of excessive glyphosate to animals and plants, it is of great importance to remove the compund effectively, that is, to decrease the concentration of phosphorus.

**Figure 1.** The chemical structure of glyphosate.

A range of conventional methods, such as adsorption and biological treatments, have been investigated for glyphosate removal. For example, nano-$CuFe_2O_4$ modified biochar [5] as an adsorbent exhibited a maximum adsorption capacity of 269.4 mg/$g^{-1}$ within 240 min (C[glyphosate] = 600 mg $L^{-1}$, 298 K, pH = 4). Nevertheless, some limits and drawbacks still exist, such as the high cost of the adsorbent and difficulties associated with its separation [6]. Meanwhile, biological treatment is a promising method to treat glyphosate-containing water. It can achieve high glyphosate removal over a wide range of glyphosate concentrations. However, this approach cannot achieve a high removal efficiency of glyphosate because of the generation of by-products such as aminomethylphosphonic acid (AMPA) or sarcosine. Furthermore, long residence times and suitable growth conditions for microorganisms are required [7–9]. Advanced oxidation processes (AOPs) have also been developed as alternative treatment technologies for glyphosate removal. AOPs have presented greater efficiency than the other two methods. For example, 96% glyphosate removal and 63% total COD removal were accomplished under experimental conditions (T: 90 °C, pH: 3–4, reaction time: 2 h, $n(H_2O_2)/n(Fe^{2+})$ = 4:1) under a conventional Fenton system [10]. Various AOPs, including Fenton oxidation, photocatalysis, electrochemical oxidation, and ozonation, are also methods for glyphosate removal in wastewater [11]. However, among AOPs, approaches based on the Fenton system are considered favorable for glyphosate degradation in view of the advantages of simple operation, no mass transfer limitation, and easy implementation as a stand-alone or hybrid system [12]. This is due to shorter contact time and the generation of fewer by-products. Using electro-Fenton [13], total organic carbon (TOC) removal percentage reached 82% under the following conditions: $Mn^{2+}$ as catalyst; Temperature: 23 °C; pH: 3; anode: Pt cylindrical mesh; cathode: carbon felt; electrolyte: 0.05 M $Na_2SO_4$. Another representative study [14] claimed that within 120 min, 75% TOC, and 0.415 mmol $L^{-1}$ of glyphosate removal and 36% toxicity decrease were achieved by photo-Fenton. To conclude, for glyphosate removal, AOP approaches are remarkably effective in comparison to adsorption and biological treatments, even though the $H_2O_2$ used in AOPs has low stability [15]. Thus far, this problem has received attention, as it is quite challenging and inconvenient to store and transport liquid hydrogen peroxide [16]. To resolve this concern, calcium peroxide ($CaO_2$), known as the solid form of hydrogen peroxide, is being considered for use as an oxidant in Fenton systems to remove various types of pollutants.

In response to a growing need to address environmental contamination in wastewater, $CaO_2$, as a solid peroxide, has been increasingly used in wastewater treatment, surface water restoration, groundwater, and soil remediation as an environmentally friendly chemical that can release oxygen and hydrogen peroxide at a controlled rate [17]. It has high thermal stability and is less affected by atmospheric moisture and carbon dioxide. It is a white or yellowish solid that belongs to the alkaline earth metal peroxide group [18]. Additionally, calcium peroxide can generate active oxidizing hydroxyl free radicals (·OH) in the presence of $Fe^{2+}$ in an acidic solution, resulting in excellent removal efficiency. The reaction mechanism is as follows:

$$CaO_2 + 2H^+ \rightarrow Ca^{2+} + H_2O_2 \tag{1}$$

$$H_2O_2 + Fe^{2+} \rightarrow Fe^{3+} + OH^- + ·OH \tag{2}$$

$$Fe^{2+} + ·OH \rightarrow Fe^{3+} + OH \tag{3}$$

$$Fe^{3+} + H_2O_2 \rightarrow Fe^{2+} + H^+ + HO_2\cdot \quad (4)$$

$$HO_2\cdot + H_2O_2 \rightarrow O_2 + H_2O + \cdot OH \quad (5)$$

In a recent study, the potential of $CaO_2$ as an advanced oxidant was demonstrated. Notably, 99.93% decolorization and 81.82% COD removal were achieved using $CaO_2/H+/Fe^{2+}$ advanced Fenton-like oxidation technology for textile wastewater treatment [19]. Similarly, it has been reported that benzene, toluene, ethylbenzene, and xylene (BTEX) in wastewater can be removed simultaneously by a $CaO_2$-based Fenton system [20].

In the present work, calcium peroxide was selected to remove glyphosate from aqueous solutions. Fourier transform infrared spectroscopy (FTIR), X-ray diffraction (XRD), and potassium permanganate titration were conducted for the characterization of calcium peroxide. The effect of pH, the molar ratio of $Ca^{2+}:Fe^{2+}$, the initial $CaO_2$ dosage, and the initial glyphosate concentration were also reported with pseudo-zero order, pseudo-first order, and pseudo-second order rate laws and the BMG model.

## 2. Materials and Methods

### 2.1. Chemical Reagents and Instruments

A 1000 ppm glyphosate stock solution was diluted from Roundup (commercial grade, containing 360 g/l glyphosate). The glyphosate solutions in varying concentrations with a natural pH of 4.9–5.4 were then prepared by diluting the stock solution with distilled water. Calcium peroxide ($CaO_2$, 65%) was purchased from Alfa Aesar, Ward Hill, MA, USA. Ferrous sulfate (FeSO4.7H2O, $\geq$98%) was obtained from R&M Chemicals Sdn. Bhd. (Semenyih, Malaysia). Sodium sulfite ($Na_2SO_3$, $\geq$98%, 2.0 M), sodium hydroxide (NaOH), and sulfuric acid ($H_2SO_4$) were supplied from Sigma Aldrich (St Louis, MO, USA) and were of analytical grade. The pH of the solution was measured using a digital pH meter (Model: Sartorius PB-10). The desired pH of the solution was adjusted by the addition of standardized 0.1 M $H_2SO_4$ and 0.1 M NaOH solutions. An ultraviolet–visible spectrophotometer (Model: Dynamica, HALO DB-20) was used to monitor the degradation process. The COD values were analyzed before and after treatment via the dichromate digestion-colorimetric method (Lovibond thermoreactor RD125, DR/890 Portable colorimeter). The FTIR spectra were obtained by a compact FTIR Spectrometer (Bruker ALPHA II) in the range of 500–4000 cm$^{-1}$ and XRD analysis was performed by X-ray diffractometer (PHILIPS PW 3040/60 MPD X' Pert high pro-PANalytical).

### 2.2. Methods

Fourier transform infrared spectroscopy (FTIR) and X-ray diffraction (XRD) were conducted, respectively, to characterize the calcium peroxide. Potassium permanganate titration was used to measure the purity of $CaO_2$ in the sample [21]. The purity was calculated using the following equation.

$$purity(\%) = \frac{5 \times C(KMnO_4) \times V(KMnO_4) \times 72.08}{2 \times m(CaO_2) \times 1000} \times 100 \quad (6)$$

where $C(KMnO_4)$ = 0.02 M. Titration was run in triplicate and the mean value was obtained.

In a typical experiment, a glyphosate-containing solution of 50 ppm was prepared in a volumetric flask with distilled water from the stock solution (1000 ppm). Then, 100 mL 50 ppm glyphosate solution was placed in a 250 mL glass conical flask with the desired amount of $CaO_2$ and $FeSO_4$. After the pH had been adjusted, all experiments were carried out in a water bath shaker at a speed of 150 rpm. To quench the Fenton reaction, the excess of 2.0 M $Na_2SO_3$ was added to the samples, ensuring the removal of the remaining $H_2O_2$ [14]. The ammonium molybdate spectrophotometry method was applied to detect total phosphorus in an aqueous solution. The absorbance of each sample was determined at $\lambda$ = 880 nm and the calibration curve of glyphosate ($R^2$ = 0.9938) was prepared. Ultimately, the total removal efficiency was calculated by the following equation:

$$total\ removel\ efficiency(\%) \ = \ \left(1 - \frac{C_t}{C_0}\right) \times 100 \tag{7}$$

where $C_0$ and $C_t$ are the concentration of total phosphate at the initial and a given contact time $t$, respectively.

COD was determined before and after treatment for each parameter, and the *COD* removal percentage was calculated using the following equation:

$$COD\ removal\ (\%) \ = \ \frac{COD_{before\ treatment} - COD_{after\ treatment}}{COD_{before\ treatment}} \times 100 \tag{8}$$

All experiments were carried out in triplicate, and the mean values were reported. A schematic diagram of experiment process on glyphosate removal is shown in Figure 2.

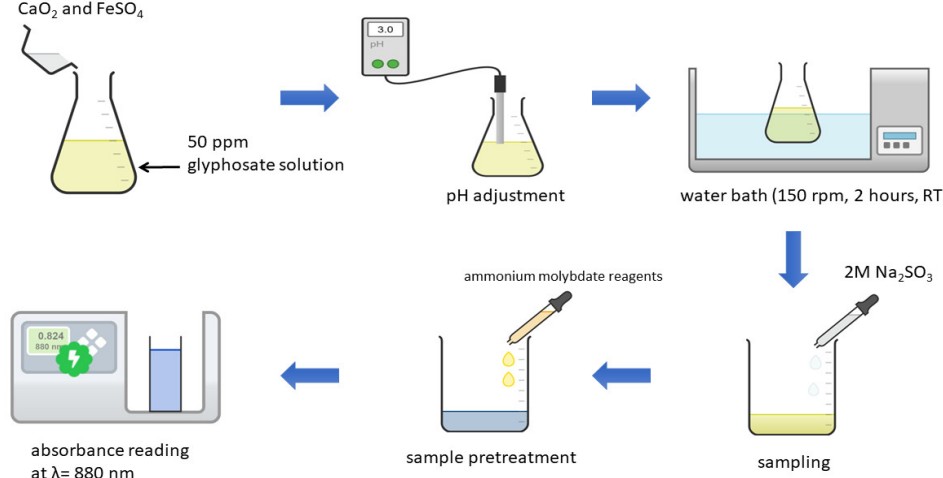

**Figure 2.** Schematic diagram of experiment process on glyphosate removal.

The following parameters were studied in the present study:

### 2.2.1. Effect of pH

In the Fenton process, the $H^+$ concentration greatly affects the formation of $H_2O_2$, which, in turn, controls the generation of the hydroxyl radicals. Thus, pH is one of the most important aspects to investigate. The Fenton reagent only works under acidic conditions because the form of Fe in the solution is limited by pH [22]. The effect of pH on glyphosate removal was evaluated at 2.0, 2.5, 3.0, 3.5, and 4.0. In this set, experiments were conducted under controlled conditions (initial $CaO_2$ dosage = 0.5 g, the molar ratio of $Ca^{2+}:Fe^{2+}$ = 6, 100 mL 50 ppm glyphosate solution, 150 rpm, RT = 25 °C, contact time = 120 min). The absorbance values were monitored at an interval of 15 min. For the first 15 min, samples were taken every 5 min.

### 2.2.2. Effect of the Molar Ratio of $Ca^{2+}:Fe^{2+}$

Another essential parameter for the Fenton process is the molar ratio of $H_2O_2:Fe^{2+}$, $CaO_2$ for the present work, which impacts the generation of the hydroxyl radicals. Under the optimal pH of 3.0 that was obtained in the first parameter, other conditions were fixed (initial $CaO_2$ dosage = 0.5 g, 100 mL 50 ppm glyphosate solution, 150 rpm, RT = 25 °C, contact time = 120 min). Fenton-based experiments were carried out to investigate the effects of the molar ratio of $Ca^{2+}:Fe^{2+}$ on glyphosate removal ranging from 1 to 10. Since the initial $CaO_2$ dosage was fixed at 0.5 g, the dosage of $Fe^{2+}$ was varied based on the ratio. The absorbance values were monitored at an interval of 15 min. For the first 15 min, samples were taken every 5 min.

### 2.2.3. Effect of Initial $CaO_2$ Dosage

Once the optimal molar ratio of $Ca^{2+}$:$Fe^{2+}$ had been determined, the initial calcium peroxide dosage was a significant parameter to study. Under the optimal pH of 3.0 and the optimal molar ratio of $Ca^{2+}$:$Fe^{2+}$ = 6 that were obtained from the first two parameters, other conditions were fixed (100 mL 50 ppm glyphosate solution, 150 rpm, RT = 25 °C, contact time = 120 min). Fenton-based experiments were carried out to investigate the effects of initial $CaO_2$ dosage on glyphosate removal ranging from 0.1 g to 0.9 g. The dosage of $Fe^{2+}$ was changed with the dosage of $CaO_2$ because the optimal molar ratio of $Ca^{2+}$:$Fe^{2+}$ was fixed at 6. The absorbance values were monitored at an interval of 15 min. For the first 15 min, samples were taken every 5 min.

### 2.2.4. Effect of the Initial Glyphosate Concentration

Under the optimal conditions determined before (pH = 3.0, initial $CaO_2$ dosage = 0.5 g, the molar ratio of $Ca^{2+}$:$Fe^{2+}$ = 6), the effect of initial glyphosate concentration was studied. The removal performance of glyphosate in solution was measured at different initial glyphosate concentrations (10, 20, 30, 40, and 50 ppm, 100 mL for each). A contact time of 30 min at room temperature was selected and the absorbance values were acquired at 0, 5, 10, 20, and 30 min.

### 2.2.5. Kinetics Study

Four kinetic models (pseudo-zero order, pseudo-first order, pseudo-second order, and BMG models) were employed in this work for the kinetics studies [15]. The modeling equations are listed in Table 1:

**Table 1.** Kinetic modelling equations.

| Order | Equation Applied | Linear Form by Integration |
|---|---|---|
| Zero-order | $\dfrac{dC_t}{d_t} = -k_0$ | $C_t = C_0 - k_0 \cdot t$ |
| First-order | $\dfrac{dC_t}{d_t} = -k_1 \cdot C_t$ | $lnC_t = lnC_0 - k_1 \cdot t$ |
| Second-order | $\dfrac{dC_t}{d_t} = -k_2 \cdot (C_t)^2$ | $\dfrac{1}{C_t} = \dfrac{1}{C_0} + k_2 \cdot t$ |
| BMG model | $\dfrac{C_1}{C_0} = 1 - \left[ \dfrac{t}{(m + b \cdot t)} \right]$ | $\dfrac{t}{\left[ 1 - \left( \dfrac{C_t}{C_0} \right) \right]} = m + b \cdot t$ |

Note: $k_0$, $k_1$, and $k_2$ are apparent kinetic rate constants of zero-, first-, and second-order models, respectively; $t$ is reaction time, and $Ct$ is the concentration at a given time $t$; where $m$ and $b$ are two constants concerning initial reaction rate and maximum oxidation capacity, respectively.

## 3. Results

### 3.1. Characterization

Figure 3 depicts the FTIR spectra of $CaO_2$ to identify the functional groups present in the sample. According to the spectra, the peaks at around 855 and 875 cm$^{-1}$ corresponded to the O–O bridge of $CaO_2$. Additionally, the peak at 590 cm$^{-1}$ and the broad peak at 1487 cm$^{-1}$ of $CaO_2$ were attributed to O–Ca–O vibrations [23]. Moreover, a sharp peak around 3640 cm$^{-1}$ representing O–H stretching band was also observed, which suggested the impurity of moisture in the sample. As shown in the XRD image (Figure 4), the sample presented dominant peaks of 2θ at 30.1, 35.6, 47.3, 57.4, and 60.7, in agreement with standard data. Table 2 illustrates the purity of $CaO_2$ measured using the standard potassium permanganate titration method. The average purity was 63.30%, which is very close to the purity stated by the supplier.

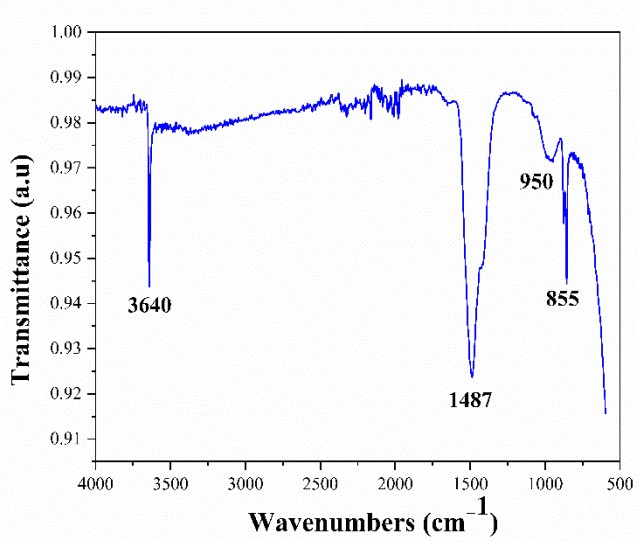

**Figure 3.** FTIR spectra of calcium peroxide.

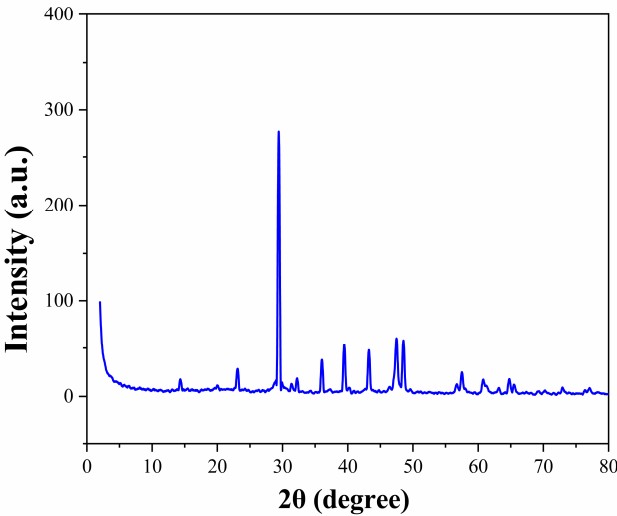

**Figure 4.** XRD patterns of calcium peroxide.

**Table 2.** Purity of calcium peroxide.

| Sample | m(CaO$_2$)/g | V(KMnO$_4$)/mL | Purity/% | Mean |
|:------:|:------------:|:--------------:|:--------:|:----:|
| 1 | 0.0571 | 10.10 | 63.75 | |
| 2 | 0.0587 | 10.25 | 62.95 | 63.30% |
| 3 | 0.0545 | 9.55 | 63.15 | |

### 3.2. Effect of pH

As the Fenton process is substantially affected by solution pH, the performance of glyphosate removal using Fenton-like system was examined at various pH values. According to our previous study, the Fenton reagent only works in acidic conditions [18]. Typically, the optimal pH lies in the range of 2 to 4 [24]. To determine the optimal pH, degradation processes at different pH conditions ranging from 2 to 4 (interval: 0.5) were investigated. It can be observed from Figure 5 that pH affects the degradation efficiency to a great extent. Concisely, the total removal efficiency within 120 min could be sequenced: pH = 3 > pH = 3.5 > pH = 2.5 > pH = 2.0 > pH = 4.0. Increasing pH from 2.0 to 3.0, the total removal efficiency of glyphosate increased considerably from 83.10% (60.21% COD removal) to 94.50% (68.60% COD removal) and peaked at pH = 3.0. The removal percentage

declined slightly at pH = 3.5, where 92.32% TP removal and 65.28% COD removal were achieved. It was also can be noticed that only 69.69% TP removal and 50.10% COD removal were achieved when the pH value was increased to 4.0. At lower pH, due to the scavenging action by $H^+$ ions and reaction between $[Fe(H_2O)]^{2+}$ ion and $H_2O_2$ at a slower rate, lower pH values resulted in a significant decrease in the number of ·OH radicals. Another cause is the stability of $H_2O_2$ at pH below 3, which is hindered by $H_3O^{2+}$, thereby preventing the generation of ·OH radicals [25]. The effectiveness of $Fe^{2+}$ catalysis deteriorated as pH increased. To conclude, both lower and higher pH levels than the optimal had a negative impact on process performance. The degradation process with kinetics modeling is given in Figure 6 and Table 3. By taking four kinetics models into account, the most appropriate model to report the removal process of glyphosate by studying the effect of pH is the first-order reaction kinetics (average $R^2$ = 0.9715), followed by the BMG model (average $R^2$ = 0.9614). Comparing $k_1$ values under different pH, it can be noticed that the rate constant ($k_1$) was the highest (0.0250) at pH = 3.0, which suggests that the degradation rate was the fastest at this pH. On the other hand, by considering parameter $1/m$ (initial degradation rate) by the BMG model, it is possible to observe that the initial degradation rate was the highest ($1/m$ = 0.0837) under pH = 3.0. Also, the initial degradation rate under pH = 3.0 was two times greater than processes under pH = 2.5 and 3.5. Due to the R square values ($R^2$ < 0.9) of zero order and second order kinetics models, the experimental data could not be fitted well. This is a vital finding in the understanding of the total removal efficiency; at pH = 3.0, the total removal percentage reached 94.5%. This result is in agreement with a previous study which claimed that the optimum pH range for a Fenton-based system was 2.5 to 3.5, differing according to the pollutant or experimental design [10].

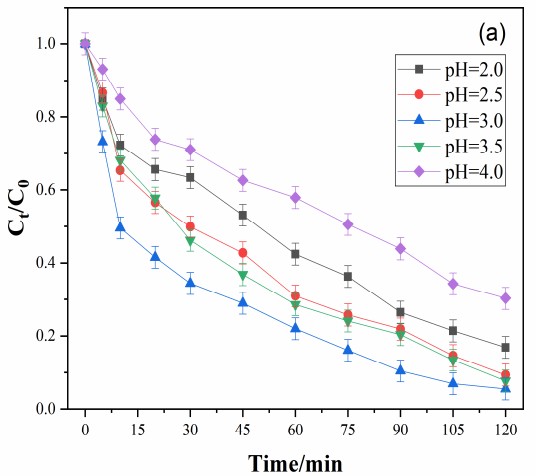 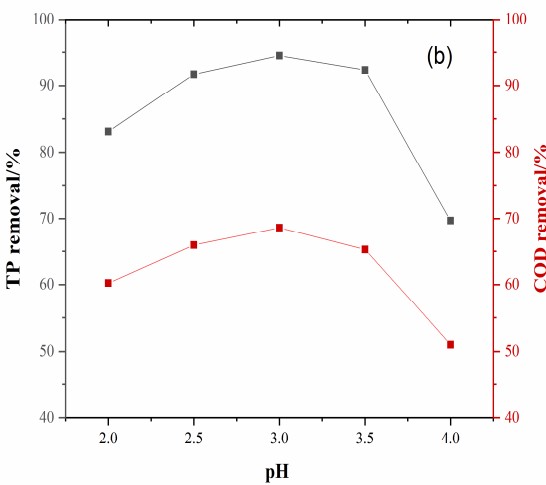

**Figure 5.** (**a**) Effect of pH on glyphosate removal; (**b**) Effect of pH on TP and COD removal (initial $CaO_2$ dosage = 0.5 g, the molar ratio of $Ca^{2+}:Fe^{2+}$ = 6, 100 mL 50 ppm glyphosate solution, 150 rpm, RT = 25 °C, contact time = 120 min).

**Table 3.** Comparison of kinetic models at different pH.

| pH | Removal (%) | Zero-Order | | First-Order | | Second-Order | | BMG Model | | |
|---|---|---|---|---|---|---|---|---|---|---|
| | | $k_0$ | $R^2$ | $k_1$ | $R^2$ | $k_2$ | $R^2$ | $1/m$ | $1/b$ | $R^2$ |
| 2.0 | 83.10 | 0.3108 | 0.9407 | 0.0138 | 0.9876 | 7.2772 | 0.9199 | 0.0327 | 0.9467 | 0.9502 |
| 2.5 | 91.68 | 0.3263 | 0.8843 | 0.0180 | 0.9725 | 0.0014 | 0.7979 | 0.0431 | 1.0185 | 0.9726 |
| 3.0 | 94.50 | 0.3139 | 0.7792 | 0.0250 | 0.9567 | 0.0026 | 0.8790 | 0.0837 | 1.0053 | 0.9919 |
| 3.5 | 92.32 | 0.3449 | 0.8894 | 0.0198 | 0.9549 | 0.0019 | 0.6753 | 0.0465 | 1.0256 | 0.9774 |
| 4.0 | 69.69 | 0.2703 | 0.9819 | 0.0093 | 0.9858 | 3.5069 | 0.9513 | 0.0191 | 0.8627 | 0.9148 |
| | Average $R^2$ | | 0.8951 | | 0.9715 | | 0.8447 | | | 0.9614 |

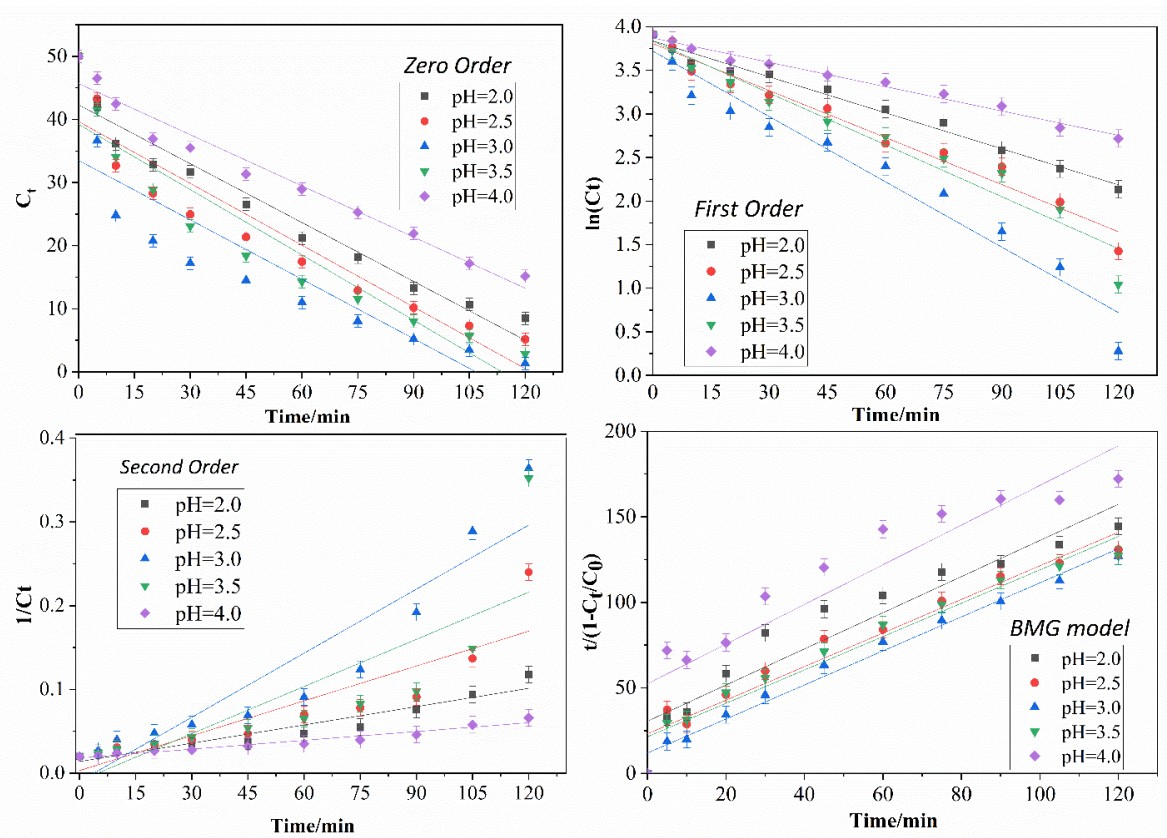

**Figure 6.** Degradation data of glyphosate using different kinetic models under different pH.

Reaction system: initial $CaO_2$ dosage = 0.5 g, the molar ratio of $Ca^{2+}:Fe^{2+}$ = 6, 100 mL 50 ppm glyphosate solution, 150 rpm, RT = 25 °C, contact time = 120 min.

### 3.3. Effect of the Molar Ratio of $Ca^{2+}:Fe^{2+}$

The ratio of Fenton reagent ($H_2O_2/Fe^{2+}$) directly impacts the treatment cost and efficiency. TP removal corresponding to different molar ratios of $Ca^{2+}:Fe^{2+}$ is revealed in Figure 7. After 120 min, TP removal percentages were 82.99, 88.60, 93.36, 94.50, 90.94, and 66.92 at molar ratio ($Ca^{2+}:Fe^{2+}$) 1, 2, 4, 6, 8, and 10, respectively. Total COD removal percentages at molar ratio ($Ca^{2+}:Fe^{2+}$) = 1–10 were 50.33, 60.88, 63.42, 68.60, 62.08, and 41.66. The results showed an increase in TP and COD removal with an increase in the molar ratio of $Ca^{2+}:Fe^{2}$ up to molar ratio = 6. However, with an increase in the molar ratio from 6 to 10, the TP removal and COD removal both dropped drastically to 66.92% and 41.66%. This was because that hydroxide radical directly reacted with metal ions with higher amounts of $Fe^{2+}$, as illustrated in Equation (3). At a higher molar ratio of $Ca^{2+}:Fe^{2+}$, the amount of $Fe^{2+}$ was smaller, which produced less ·OH radicals, leading to the lower removal efficiency [6]. Thus, it can be concluded that a molar ratio of $Ca^{2+}:Fe^{2+}$ = 6 was the most effective for the system within 120 min. The kinetics data are given in Figure 8 and Table 4. By comparing the average correlation coefficient values of four models, the BMG model fitted the degradation process most due to average $R^2$ = 0.9807, which was slightly higher and more reliable than the first-order model ($R^2$ = 0.9737). Considering parameter $1/m$ (initial degradation rate) by the BMG model, it can be seen that the initial degradation rate was the highest ($1/m$ = 0.1406) under the molar ratio of $Ca^{2+}:Fe^{2+}$ = 6. This value is significantly greater than experimental data under other ratios, indicating that the glyphosate removal can be much more effective when a molar ratio of $Ca^{2+}:Fe^{2+}$ = 6 is used. By comparing maximum oxidation capacity values ($1/b$), there was no significant change, especially for ratio = 4 or 6. This can be verified from the total removal percentages under ratio = 4 (92.36%) and ratio = 6 (94.50%). When it came to the rate constant $k_1$, the

highest value was 0.0250 at ratio = 6, while the lowest was 0.0082 at ratio = 10, which was consistent with the total removal efficiency. There was no fitting of experimental data when using the zero-order model due to the R square values ($R^2$ = 0.8152 only).

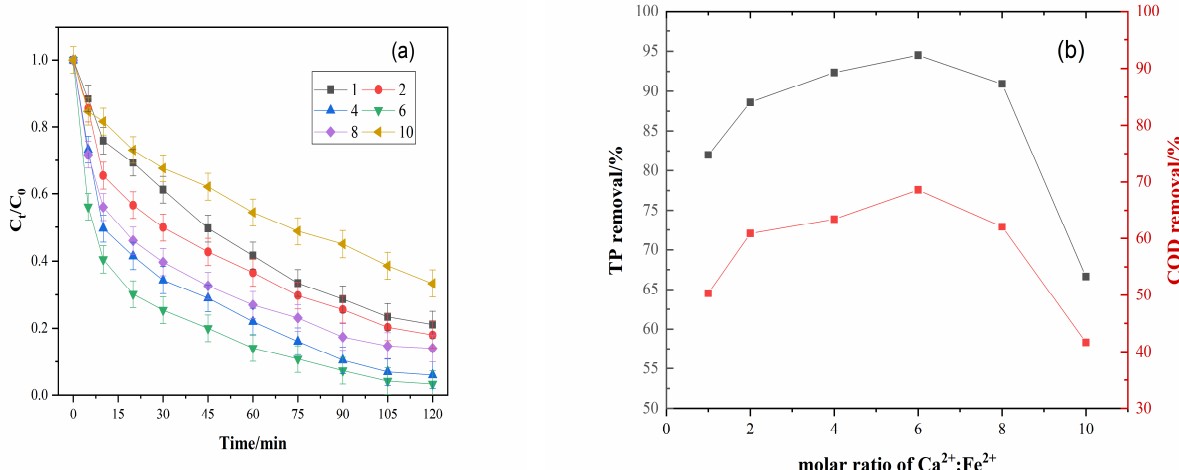

**Figure 7.** (**a**) Effect of the molar ratio of $Ca^{2+}$:$Fe^{2+}$ on glyphosate removal; (**b**) Effect of the molar ratio of $Ca^{2+}$:$Fe^{2+}$ on TP and COD removal (initial $CaO_2$ dosage = 0.5 g, 100 mL 50 ppm glyphosate solution, 150 rpm, pH = 3.0, RT = 25 °C, contact time = 120 min).

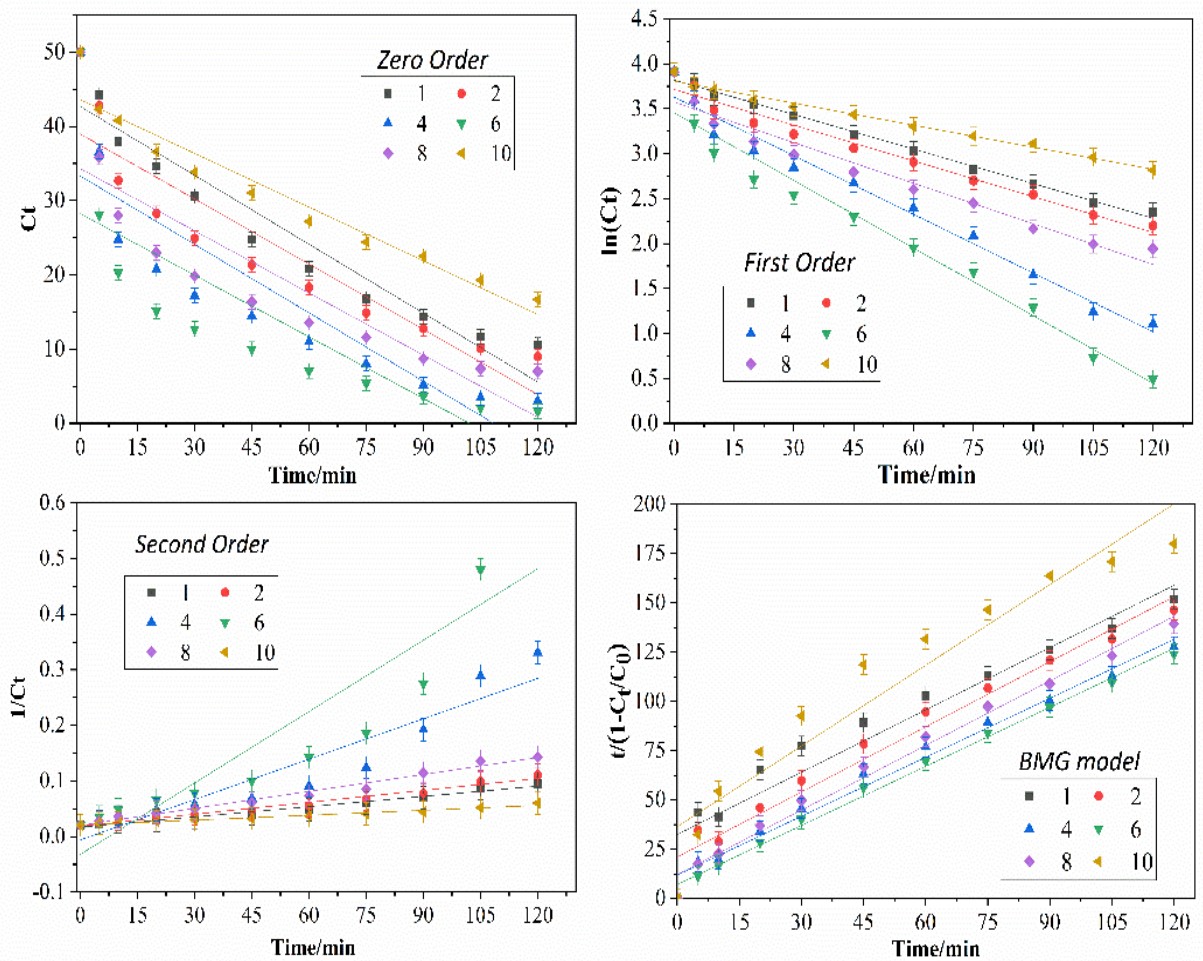

**Figure 8.** Degradation data of glyphosate using different kinetic models under different molar ratios of $Ca^{2+}$:$Fe^{2+}$.

Reaction system: initial $CaO_2$ dosage = 0.5 g, 100 mL 50 ppm glyphosate solution, 150 rpm, pH = 3.0, RT = 25 °C, contact time = 120 min.

**Table 4.** Comparison of kinetic models at different molar ratios of $Ca^{2+}:Fe^{2+}$.

| Ratio | Removal (%) | Zero-Order | | First-Order | | Second-Order | | BMG Model | | |
|---|---|---|---|---|---|---|---|---|---|---|
| | | $k_0$ | $R^2$ | $k_1$ | $R^2$ | $k_2$ | $R^2$ | $1/m$ | $1/b$ | $R^2$ |
| 1 | 81.99 | 0.3095 | 0.9245 | 0.0128 | 0.9926 | 6.1462 | 0.9794 | 0.0308 | 0.9489 | 0.9610 |
| 2 | 88.60 | 0.2924 | 0.8480 | 0.0132 | 0.9715 | 7.1500 | 0.9768 | 0.0475 | 0.9078 | 0.9827 |
| 4 | 92.36 | 0.3037 | 0.7645 | 0.0217 | 0.9764 | 0.0024 | 0.8959 | 0.0843 | 1.0025 | 0.9922 |
| 6 | 94.50 | 0.2766 | 0.6518 | 0.0250 | 0.9687 | 0.0043 | 0.8556 | 0.1406 | 1.0005 | 0.9970 |
| 8 | 90.94 | 0.2792 | 0.7626 | 0.0150 | 0.9493 | 0.0010 | 0.9825 | 0.0824 | 0.9152 | 0.9936 |
| 10 | 66.62 | 0.2411 | 0.9399 | 0.0082 | 0.9836 | 2.9442 | 0.9736 | 0.0275 | 0.7322 | 0.9576 |
| Average $R^2$ | | | 0.8152 | | 0.9737 | | 0.9440 | | | 0.9807 |

### 3.4. Effect of Initial $CaO_2$ Dosage

Glyphosate removal processes with different $CaO_2$ dosages (0.1 g, 0.3 g, 0.5 g, 0.7 g, and 0.9 g) at pH = 3.0 within 120 min at room temperature were investigated, as shown in Figure 9. A dramatic increase in TP removal appeared, from 55.08.% (0.1 g $CaO_2$) to 94.50% (0.5 g $CaO_2$). Nonetheless, as the initial $CaO_2$ dosage increased to 0.9 g, the TP removal percentage dropped to 85.15%, which implied that an excess of $CaO_2$ negatively affects degradation. This is because excessive $CaO_2$ can emit an excessive amount of $H_2O_2$, which can scavenger the Fenton reaction [26]. As shown in Figure 9 (b), the COD removal efficiency increased markedly from 17.36% at dosage = 0.1 g to 43.26% at dosage = 0.3 g. It peaked at 68.60%, i.e., the point (initial $CaO_2$ dosage = 0.5 g) where the highest TP removal (94.50%) was observed. Increasing the dosage from 0.5 g to 0.9 g, 61.01%, and 53.81% COD removal was noted, together with a moderate decrease in TP removal. The kinetics data are displayed in Figure 10 and Table 5. As seen in Table 5, the BMG model fitted even better to the experimental data, as the average $R^2$ value of the BMG model is clearly higher ($R^2 = 0.9942$) than zero-, first-, second-order kinetics models. Regarding parameter $1/m$ (initial degradation rate) by the BMG model, it is possible to observe that the initial degradation rate was the highest ($1/m = 0.1493$) when the initial $CaO_2$ dosage was 0.5 g. Considering the values of $1/b$, when the initial $CaO_2$ dosage was 0.5 g, the maximum oxidation capacity ($1/b = 1.0058$) was achieved. It can be also noted that average $R^2$ values were very close, i.e., 0.9717 for the first-order and 0.9633 for the second-order. At the optimal dosage (0.5 g $CaO_2$), k1 is better able to indicate the degradation rate. Similarly, due to the R square values ($R^2 = 0.7404$) of the zero-order kinetics model, the experimental data could not be fitted.

**Table 5.** Comparison of kinetic models at different initial $CaO_2$ dosages.

| Initial $CaO_2$ Dosage/g | Removal (%) | Zero-Order | | First-Order | | Second-Order | | BMG Model | | |
|---|---|---|---|---|---|---|---|---|---|---|
| | | $k_0$ | $R^2$ | $k_1$ | $R^2$ | $k_2$ | $R^2$ | $1/m$ | $1/b$ | $R^2$ |
| 0.1 | 55.08 | 0.1862 | 0.7638 | 0.0128 | 0.9926 | 1.9050 | 0.9197 | 0.0457 | 0.5901 | 0.9932 |
| 0.3 | 76.81 | 0.2656 | 0.8116 | 0.0132 | 0.9715 | 5.1894 | 0.9928 | 0.0553 | 0.8291 | 0.9904 |
| 0.5 | 94.50 | 0.2788 | 0.6425 | 0.0217 | 0.9764 | 0.0045 | 0.9146 | 0.1493 | 1.0058 | 0.9977 |
| 0.7 | 89.82 | 0.2828 | 0.6966 | 0.0205 | 0.9687 | 0.0015 | 0.9940 | 0.1106 | 0.8693 | 0.9970 |
| 0.9 | 85.15 | 0.2858 | 0.7873 | 0.0150 | 0.9493 | 9.5876 | 0.9953 | 0.0747 | 0.8404 | 0.9926 |
| Average $R^2$ | | | 0.7404 | | 0.9717 | | 0.9633 | | | 0.9942 |

Reaction system: the molar ratio of $Ca^{2+}:Fe^{2+} = 6$, 100 mL 50 ppm glyphosate solution, 150 rpm, pH = 3.0, RT = 25 °C, contact time = 120 min.

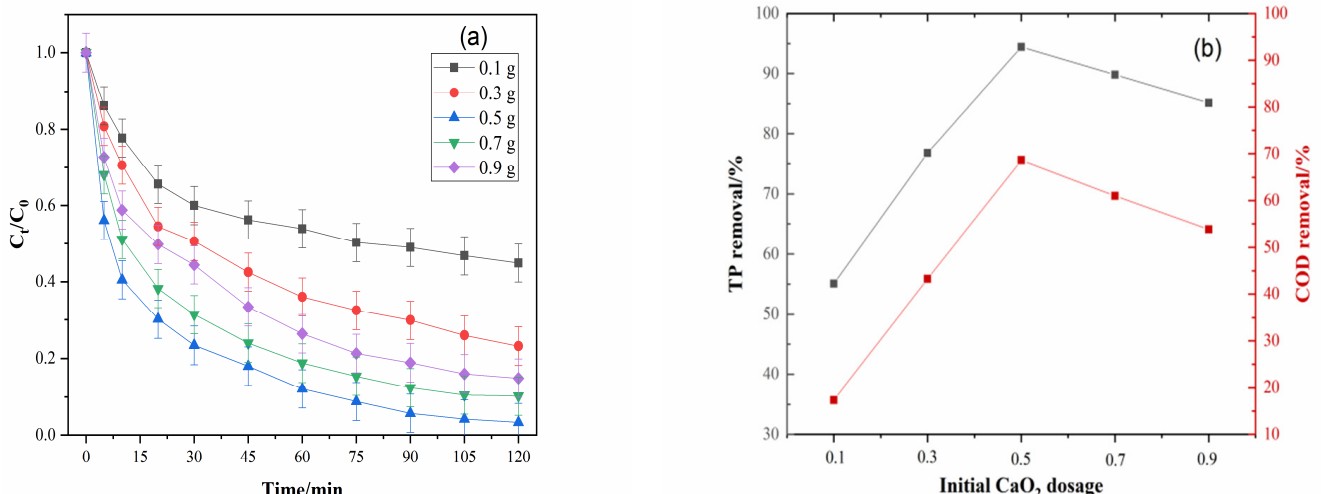

**Figure 9.** (**a**) Effect of initial $CaO_2$ dosage on glyphosate removal; (**b**) Effect of initial $CaO_2$ dosage on TP and COD removal (the molar ratio of $Ca^{2+}$:$Fe^{2+}$ = 6, 100 mL 50 ppm glyphosate solution, 150 rpm, pH = 3.0, RT = 25 °C, contact time = 120 min).

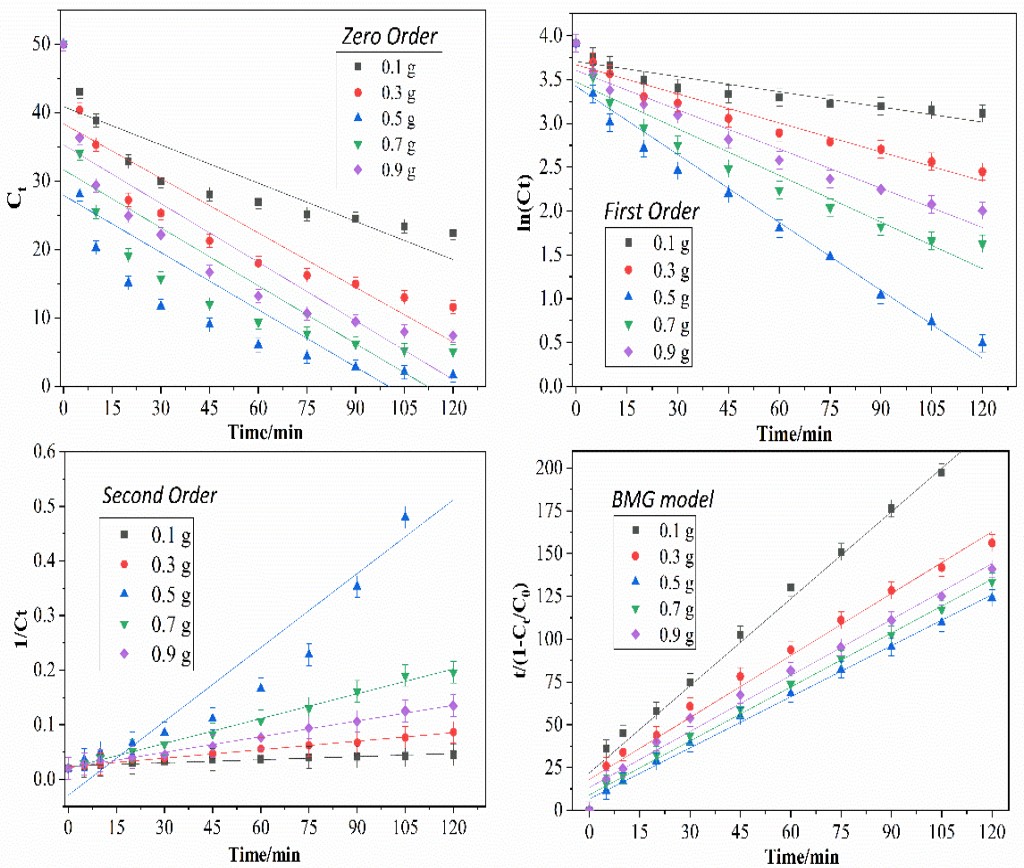

**Figure 10.** Degradation data of glyphosate using different kinetic models under different initial $CaO_2$ dosages.

## 3.5. Effect of the Initial Glyphosate Concentration

The effect of the initial glyphosate concentration on TP and COD removal was studied for a duration of 30 min. The total removal performance over 30 min is shown in Figure 11. TP removal reduced from 99.67% (85.34% COD removal) to 74.1% (50.60% COD removal) as the initial glyphosate concentration was increased from 10 ppm to 50 ppm under the optimal conditions (pH = 3.0, initial $CaO_2$ dosage = 0.5 g, the molar ratio of $Ca^{2+}$:$Fe^{2+}$ = 6,

150 rpm, room temperature). Since the oxidant ($CaO_2$) dosage was the same (0.5 g) for each experiment, the same amount of free radicals OH· were generated in all cases. The highest TP removal at 99.67% with 85.34% COD removal was achieved at the initial glyphosate concentration = 10 ppm, from which it can be concluded that the removal efficiency was significantly restricted by the amount of OH· free radicals produced from $CaO_2$. The fitted modeling data are shown in Figure 12 and Table 6. According to the correlation coefficient values, the zero-order fitted better to the experimental data, as it obtained higher average $R^2$ ($R^2 = 0.9942$) values than first-order kinetic model ($R^2 = 0.9338$), second-order kinetic model ($R^2 = 0.9419$), and the BMG kinetic model ($R^2 = 0.9658$). Considering parameter $1/m$ (initial degradation rate) by the BMG model, it can be seen that the initial degradation rate under initial glyphosate concentration = 10 ppm is the highest ($1/m = 0.5905$). This value is approximately 12–20 times higher than experimental data under other initial glyphosate concentrations, showing the more effective removal performance in the initial stage. By comparing the maximum oxidation capacity values ($1/b$), there was no significant increase or decrease at the stage of the first 30 min, indicating that the maximum oxidation capacity for every parameter was similar. From the literature [27], the removal of contaminants by Fenton processes using $Fe^{2+}$ as a catalyst usually proceeds through two stages: a fast one and a much slower one. The fast stage at the beginning of the Fenton reaction is attributed to the reaction between $Fe^{2+}$ and $H_2O_2$ [28]. Therefore, the conclusion that can be drawn for studying this parameter is that at the first stage of glyphosate removal, zero-order was selected as the fitted model because of the highest $R^2$ value. However, due to the values of $1/m$ and $1/b$, the BMG model is better suited to describing reaction kinetics.

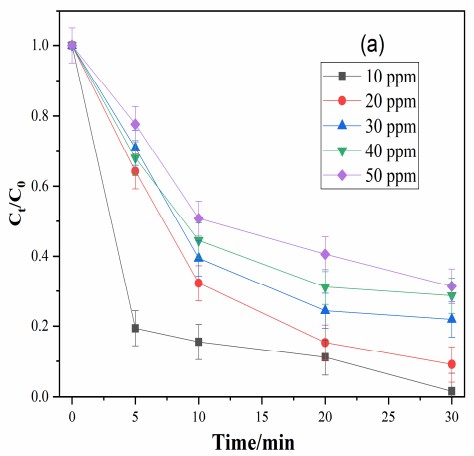
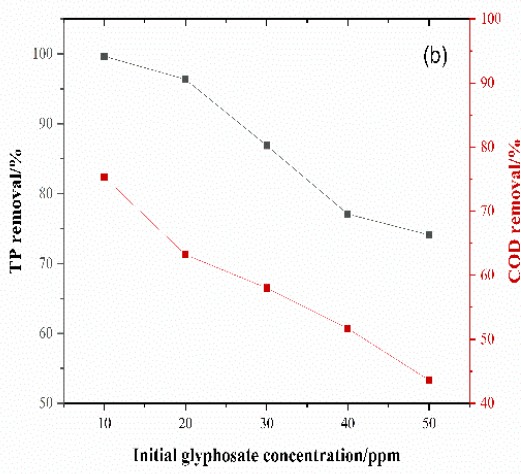

**Figure 11.** (**a**) Effect of the initial glyphosate concentration on glyphosate removal; (**b**) Effect of the initial glyphosate concentration on TP and COD removal (initial $CaO_2$ dosage = 0.5 g, the molar ratio of $Ca^{2+}:Fe^{2+} = 6$, 150 rpm, pH = 3.0, RT = 25 °C, contact time = 30 min).

**Table 6.** Comparison of kinetic models at different initial glyphosate concentrations.

| Initial Glyphosate Concentration/ppm | Removal (%) | Zero-Order | | First-Order | | Second-Order | | BMG Model | | |
|---|---|---|---|---|---|---|---|---|---|---|
| | | $k_0$ | $R^2$ | $k_1$ | $R^2$ | $k_2$ | $R^2$ | $1/m$ | $1/b$ | $R^2$ |
| 10 | 99.67 | 0.2407 | 0.5264 | 0.1142 | 0.8862 | 0.0899 | 0.8229 | 0.5905 | 0.9817 | 0.9969 |
| 20 | 96.35 | 0.5687 | 0.8190 | 0.0813 | 0.9720 | 0.0170 | 0.9788 | 0.0530 | 1.0172 | 0.9641 |
| 30 | 86.88 | 0.7571 | 0.9070 | 0.0509 | 0.9842 | 0.0039 | 0.9812 | 0.0321 | 0.8913 | 0.9553 |
| 40 | 77.07 | 0.8981 | 0.8714 | 0.0399 | 0.9569 | 0.0020 | 0.9840 | 0.0499 | 0.7845 | 0.9769 |
| 50 | 74.10 | 0.9768 | 0.9374 | 0.0355 | 0.8701 | 0.0014 | 0.9426 | 0.0168 | 0.7851 | 0.9357 |
| Average $R^2$ | | | 0.8122 | | 0.9338 | | 0.9419 | | | 0.9658 |

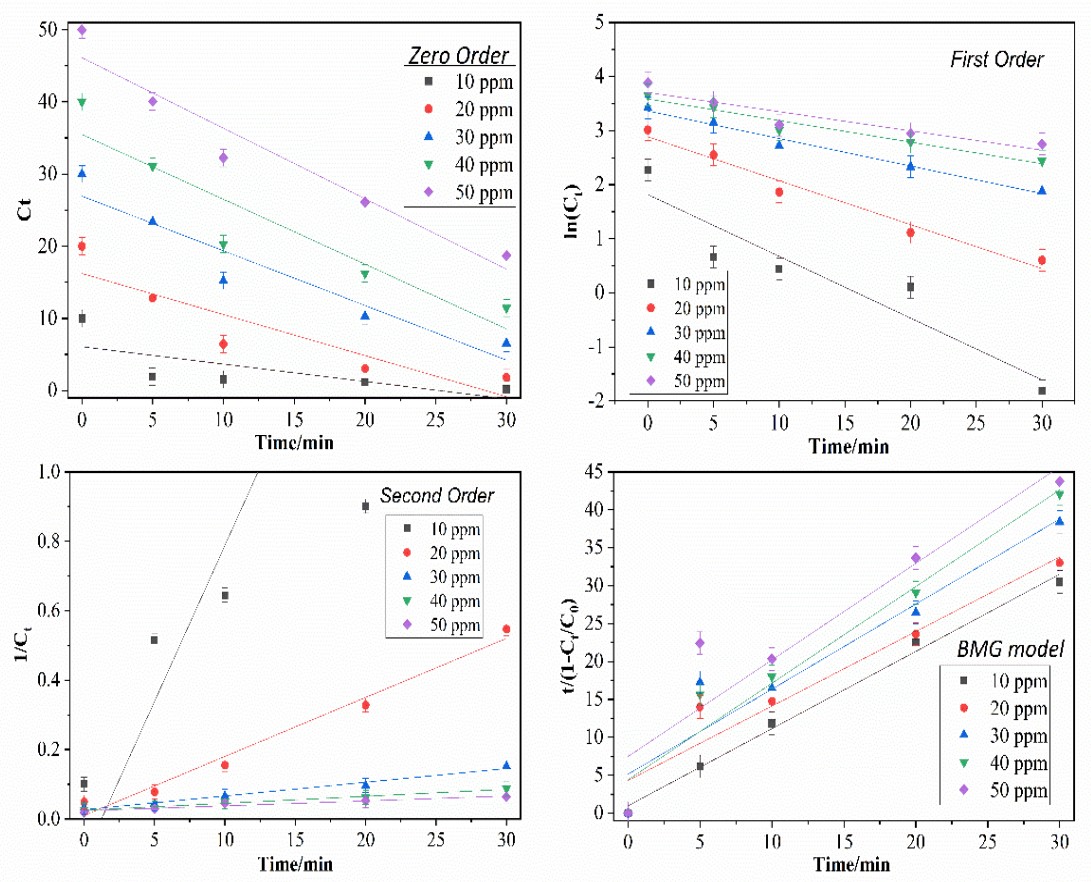

**Figure 12.** Degradation data of glyphosate using different kinetic models under different initial glyphosate concentrations.

Reaction system: initial $CaO_2$ dosage = 0.5 g, the molar ratio of $Ca^{2+}:Fe^{2+}$ = 6, 150 rpm, pH = 3.0, RT = 25 °C, contact time = 30 min.

## 4. Conclusions

The present work investigated the performance of calcium peroxide for glyphosate removal using a Fenton-like system comprised of $CaO_2$ in the presence of a $Fe^{2+}$ catalyst. The optimal operating conditions were as follows: pH = 3.0, the initial glyphosate concentration = 50 ppm, initial $CaO_2$ dosage = 0.5 g, the molar ratio of $Ca^{2+}:Fe^{2+}$ = 6, and room temperature. Under the given conditions, up to 94.50% TP removal and 68.60% COD removal were achieved within 120 min, which could be mostly fitted with the BMG kinetics model. The obtained results indicate that $CaO_2$ is an effective oxidant for glyphosate removal. Additionally, pH, the molar ratio of $Ca^{2+}:Fe^{2+}$ and initial calcium peroxide dosage have a significant impact on the glyphosate removal.

**Author Contributions:** Conceptualization, F.L., T.S.Y.C., S.S., L.C.A., S.N.A.M.J. and N.N.A.N.; methodology, F.L., T.S.Y.C., S.S., L.C.A., S.N.A.M.J. and N.N.A.N.; software, F.L., T.S.Y.C., S.S., L.C.A., S.N.A.M.J. and N.N.A.N.; validation, F.L., T.S.Y.C., S.S., L.C.A., S.N.A.M.J. and N.N.A.N.; formal analysis, F.L., T.S.Y.C., S.S., L.C.A., S.N.A.M.J. and N.N.A.N.; investigation, F.L., T.S.Y.C., S.S., L.C.A., S.N.A.M.J. and N.N.A.N.; resources, F.L., T.S.Y.C., S.S., L.C.A., S.N.A.M.J. and N.N.A.N.; data curation, F.L., T.S.Y.C., S.S., L.C.A., S.N.A.M.J. and N.N.A.N.; writing—original draft preparation, F.L., T.S.Y.C., S.S., L.C.A., S.N.A.M.J. and N.N.A.N.; writing—review and editing, F.L., T.S.Y.C., S.S., L.C.A., S.N.A.M.J. and N.N.A.N.; visualization, F.L., T.S.Y.C., S.S., L.C.A., S.N.A.M.J. and N.N.A.N.; supervision, F.L., T.S.Y.C., S.S., L.C.A., S.N.A.M.J. and N.N.A.N.; project administration, F.L., T.S.Y.C., S.S., L.C.A., S.N.A.M.J. and N.N.A.N.; funding acquisition, F.L., T.S.Y.C., S.S., L.C.A., S.N.A.M.J. and N.N.A.N. All authors have read and agreed to the published version of the manuscript.

**Funding:** This research was funded by the Ministry of Higher Education (KPT), Malaysia under the Fundamental Research Grant Scheme FRGS/1/2020/TK0/UPM/01/2 (03 01 20 2250FR).

**Data Availability Statement:** Data are available on demand from the first author.

**Conflicts of Interest:** No conflict of interest exists in the submission of this manuscript and all authors approve this manuscript for publication. The authors confirm here that this manuscript has not been previously published in whole and is not under consideration by any other journals.

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
