# Peer review of "Investigation of Glyphosate Removal from Aqueous Solutions Using Fenton-like System Based on Calcium Peroxide"

_processes, doi:10.3390/pr10102045_

Round 1
Reviewer 1 Report
The authors investigated the process of glyphosate removal from aqueous solutions by calcium peroxide. This method has certain practical value in the field of water environment protection. Before publication, the authors are advised to make the following revisions: 1. Photocatalysis is a relatively common method. What is the progress of this paper in terms of effect, efficiency, and cost compared with the method? 2. The experimental process and results of this paper are scientific, and some pictures can be further optimized. For example, the horizontal dimensions of Figure 3 and Figure 4 can be further utilized. 3. What is the most efficient reaction temperature for this method?
Reviewer 2 Report
Reformulate "Glyphosate, an organophosphate herbicide containing carboxyl group, is widely used in agriculture." the information doesn't seem correct.
observing that Fe2+ is used in the manuscript, i suggest the authors to change the title because is inadequate. please justify the role Fe2+ in the manuscript.
I suggest the authors to redo the abstract and make connection between phrases.
Reformulate : " Under this circumstance, agricultural wastewater caused by the extensive use of chemical fertilizers, and pesticides and herbicides should be stressed"
add please reference where is needed, by example for the reaction at row 93, but there are many places missing the references
add please the manufactured of the reagent
you add beside CaO2 the FeSO4 why did you do the FTIR just for CaO2?
row 211 - CaO2/Fe2+/glyphosate ? is a new material ? why did you not done ATR for solutions?
row 366-367 - CaO2 ? or Ca2+: Fe2+ which one presents the best results ?
Round 2
Reviewer 2 Report
The authors have improved the manuscript.